# Changes in Water Soluble Uremic Toxins and Urinary Acute Kidney Injury Biomarkers After 10- and 100-km Runs

**DOI:** 10.3390/ijerph16214153

**Published:** 2019-10-28

**Authors:** Wojciech Wołyniec, Katarzyna Kasprowicz, Joanna Giebułtowicz, Natalia Korytowska, Katarzyna Zorena, Maria Bartoszewicz, Patrycja Rita-Tkachenko, Marcin Renke, Wojciech Ratkowski

**Affiliations:** 1Department of Occupational, Metabolic and Internal Medicine, Institute of Maritime and Tropical Medicine, Medical University of Gdańsk, 81-519 Gdynia, Poland; mrenke@gumed.edu.pl; 2Department of Biology, Ecology and Sports Medicine, Gdańsk University of Physical Education and Sport, 80-336 Gdańsk, Poland; katarzyna.kasprowicz@awf.gda.pl; 3Department of Bioanalysis and Drug Analysis, Faculty of Pharmacy, Medical University of Warsaw, 02-097 Warsaw, Polandnatalia.korytowska@wum.edu.pl (N.K.); 4Department of Immunobiology and Environment Microbiology, Medical University of Gdańsk, 81-519 Gdynia, Poland; kzorena@gumed.edu.pl (K.Z.); maria.bartoszewicz@gumed.edu.pl (M.B.); 5Medical Laboratories Bruss Group ALAB, Gdynia, Poland 81-519 Gdynia, Poland; patrycja.rita@lmbruss.pl; 6Department of Athletics, Department of Athletics, Gdańsk University of Physical Education and Sport; 80-336 Gdańsk, Poland; maraton1954@o2.pl

**Keywords:** urinary biomarkers, uremic toxins, albuminuria, fractional excretion, SDMA, TMAO, ultramarathon

## Abstract

Acute kidney injury (AKI) is described as a relatively common complication of exercise. In clinical practice the diagnosis of AKI is based on serum creatinine, the level of which is dependent not only on glomerular filtration rate but also on muscle mass and injury. Therefore, the diagnosis of AKI is overestimated after physical exercise. The aim of this study was to determine changes in uremic toxins: creatinine, urea, uric acid, asymmetric dimethylarginine (ADMA), symmetric dimethylarginine (SDMA), trimethylamine N-oxide (TMAO) and urinary makers of AKI: albumin, neutrophil gelatinase-associated lipocalin (uNGAL), kidney injury molecule-1 and cystatin-C (uCyst-C) after long runs. Sixteen runners, mean age 36.7 ± 8.2 years, (2 women, 14 men) participating in 10- and 100-km races were studied. Blood and urine were taken before and after the races to assess markers of AKI. A statistically significant increase in creatinine, urea, uric acid, SDMA and all studied urinary AKI markers was observed. TMAO and ADMA levels did not change. The changes in studied markers seem to be a physiological reaction, because they were observed almost in every runner. The diagnosis of kidney failure after exercise is challenging. The most valuable novel markers which can help in post-exercise AKI diagnosis are uCyst-C and uNGAL.

## 1. Introduction

Acute kidney injury (AKI) is described as a relatively common complication after marathon run and other prolonged exercises [1]. The proper diagnosis of AKI is important, because repeated episodes of AKI may lead to chronic kidney disease (CKD) [2]. There is no evidence that sport practicing can lead to chronic kidney problems, but it seems to be a real danger because nowadays some runners present rather unhealthy behaviors—such as running a marathon every week [3]. The possible factors causing post-exercise AKI are dehydration, sub-clinical rhabdomyolysis, inflammation, increased energy demanding renal sodium uptake, reduced renal perfusion and nonsteroidal anti-inflammatory drugs (NSAIDs) frequently used by runners [4,5]. There is also growing evidence that dehydration or re-hydration with soft drinks (i.e., a high-fructose, caffeinated beverage) during and following exercise may lead to kidney dysfunction [6]. 

In clinical practice, the diagnosis of AKI is based only on changes in creatinine level, according to Kidney Disease Improving Global Outcomes (KDIGO) criteria [7]. This approach has some important limitations in exercising subjects. Creatinine is a breakdown product of creatine phosphate in muscles. Its serum level depends on muscle mass, diet, age, sex, and hydration status [8,9]. Moreover, rigorous exercise increases creatinine levels in serum by increasing muscle breakdown. Thus, creatinine increment after exercise reflects rather muscle injury than AKI. Therefore, the rate of AKI diagnosis after marathon can be overestimated. On the other hand, diagnosis of AKI based on creatinine level is delayed, because creatinine reaches the highest value in blood 24–48 h after injury [8]. Average amateur marathon run lasts 3–6 h and ultramarathon 10–18 h (depending on distance and weather conditions). Therefore, creatinine level measured on finish line can be underestimated, and diagnosis of severe AKI e.g. due to rhabdomyolysis, can be delayed [10]. 

Another problem is that the baseline values of creatinine, before kidney injury, are unknown. So-called back calculation is not precise and seems to be highly inadequate in exercising persons [11]. Therefore, diagnosis of AKI cannot be based on relative changes of creatinine level, which is the most rational and consistent with KDIGO [7]. The second criterium used to diagnose AKI according to AKIN is decreased urine output. There are two main problems with diagnosis of AKI based on urine output in exercising subjects. First, it is very difficult to precisely assess urine output during ultramarathon. Subject running 100km in forest are urinating several times in different places. There is no chance to collect urine in such a race. Second, according to AKIN the prolog, at least 6 h, observation is needed. It could be easily done in hospital, but it is very difficult to observe marathoners who completed 100 km run after finish line for so long. Therefore in most studies a diagnosis of AKI after ultramarathon is based only on laboratory tests [7]. 

The other “classical” uremic toxins measured in serum in AKI have similar limitations. Urea and uric acid reflect not only AKI but also increased metabolism, dehydration and hormonal activation during exercise [12]. Some new toxins, like asymmetric dimethylarginine (ADMA), symmetric dimethylarginine (SDMA) and trimethylamine N-oxide (TMAO) have been studied in kidney diseases, mainly in chronic kidney disease (CKD) [13,14]. The problem is that their levels are increased not only in CKD. ADMA, SDMA and TMAO are increased also in metabolic and cardiovascular disorders [15]. These new uremic toxins have been investigated in few studies after exercise, mainly in patients with increased cardiovascular risk [16,17]. 

The best known urinary marker of kidney dysfunction is albuminuria, which is present in acute and chronic kidney disease. In health, kidneys work very precisely to purify blood and maintain internal homeostasis and produce urine without proteins, glucose and other valuable substances [18]. This precise mechanism is changed in AKI. Any dysfunction of the kidneys has impact on urine contents. One of the first observations in nephrology was done by Bright who found not only urea retention but also presence of albumin in urine in patients with renal failure [19]. After 100 years, urinalysis is still essential part of kidney function evaluation. In exercise, albuminuria is probably a benign, although not fully understood phenomenon [20]. In diagnosis of early kidney dysfunction, several concepts appeared: the urinary excretion of enzymes, urinary proteomics and metabolomics [21,22]. So-called new biomarkers of AKI can also be studied in urine, and they are an attractive alternative to creatinine level and albuminuria [8,23]. Some of these markers are practically absent in urine of healthy people; therefore, any increase can be diagnostic. The novel urinary markers of AKI are: neutrophil gelatinase-associated lipocalin (NGAL), kidney injury molecule 1 (KIM-1), transforming growth factor beta-1, retionol-binding protein, cystatin C (Cyst-C), interleukin 18, liver-fatty-acid-binding protein, uromodulin, clusterin, and trefoil factor 3 [9,24]. 

The AKI diagnosis after intensive exercise is still a challenge. It is due to physiological increase of AKI marker—creatinine. Thus, other AKI markers are needed. In order to propose other diagnosis scheme the influence of sport on renal function need to be better understood. There are some data on changes of urinary AKI markers after prolonged exercises. However, the studies concentrate only on one marker [25,26] or few markers but during one type of exercise [27]. There is also no data on changes in novel uremic toxins such as TMAO, ADMA and SDMA in plasma of healthy people after exercises. These toxins reveal significant influence on cardiovascular system and are independent risk factors of cardiovascular disease [28,29]. Moreover, no comparison of two similar physical activities of different length was made. Thus, we propose the complex study where classical AKI markers i.e., creatinine, uric acid, urea, novel uremic toxins i.e. TMAO, ADMA, SDMA were determined in serum and novel urinary markers of AKI i.e., NGAL, KIM-1 and Cyst-C were determined in urine of volunteers after 10- and 100-km runs. It could help to assess whether changes in AKI markers in blood are the indicator of adoptive physiological state or pathological changes. 

## 2. Materials and Methods 

The runners were recruited via an invitation email sent to runners who were ready to participate in a 100-km county run. Twenty runners (18 males (M), 2 females (F), age 37.2 ± 7.4 years) responded and were included in the first part of the study. All participants of the study were healthy amateur runners without previous history of sports career. 

### 2.1. Subjects Selection 

The inclusion criteria were: readiness to participate, signed written informed consent, age between 25 and 60 years. The exclusion criteria were: a history of kidney disease, serum creatinine above 1.2 mg/dL, eGFR < 60 mL/min/1.73 m^2^ and albumin-to-creatinine ratio (ACR) > 15 mg/g or hematuria in urinalysis before the start. 

### 2.2. Races 

The first event was a 100-km county trail run. The run took place in the forest in September 2018. The official name of the event was “Kaszubska poniewierka” [30]. One subject abandoned the race after 30 km and samples were not taken. In another runner, who successfully completed the 100-km race, blood was not taken after the race due to technical problems. Both runners were excluded from the study. Complete blood and urine analysis after this race was possible in 18 runners (two F, 16 M), mean age 37.0 ± 7.8 years. Sixteen subjects completed the race, two others ran a distance of 51 km. 

The runners participated in the first study were invited to take part in 10-km field run. This race was organized individually for each participant. All 10-km races took place within 6 month after the first (100-km) race in February and March 2019. The results of 16 runners taking part in this race (2F, 14 M, mean age 36.8 ± 8.2 years) were analyzed (Figure 1). Thus, after both races, the analysis of blood and urine was possible in 16 runners with mean age 36.8 ± 8.2 years (2 F, 14 M). All the runners were of Caucasian race.

### 2.3. Questionnaires and Anthropometric Data 

Before the first race a questionnaire concerning running experience was sent to all participants. The study was preceded by noninvasive anthropometric analysis. The following parameters were estimated: heart rate, weight, height, body mass index, waist-to-hip ratio, and percent body fat. Body composition was assessed by bioelectrical impedance analysis, with a commercially available body analyzer (InBody 720; Biospace, Seoul, Korea).

### 2.4. Weather Conditions

The 100-km run took place in September. At the beginning of the race the temperature was 8 °C and then it increased to 16 °C at the end. There was no rain. The 10-km runs were organized in February and March when the mean temperature fluctuate between 2 and 5 °C. All the races were organized in the midday of rainless days.

### 2.5. Biochemical Analyses

Blood and urine samples were collected before and after the races. Blood was drawn from the antecubital vein in a sitting position by experienced nurses. The blood was centrifuged at 1000 G for 10 min. Serum and urine were frozen up and stored at −80 °C for up to 6 months before analysis.

#### 2.5.1. Urinalysis 

Before and after both races the urine was analyzed by using a ten-patch test strip for the semi-quantitative determination of specific gravity, pH, leukocytes, nitrite, protein, glucose, ketone bodies, urobilinogen, bilirubin, and blood (Cobas 411 analyzer, Roche Diagnostics GmbH, Mannheim, Germany). Elements contained in urine were measured using the flow cytometry method (UF 100i, SYSMEX EUROPE GmbH, Norderstedt, Germany).

#### 2.5.2. Measurement of Creatinine, Urea and Uric Acid

##### Creatinine, Urea and Uric Acid Were Measured in Serum and Spot Urine

Due to technical reasons, creatinine, urea, and uric acid were measured after 10- and 100-km races using different methods in the same laboratory. 

##### 100-km—Creatinine, Urea and Uric Acid Were Measured in Serum and Spot Urine

Serum and urinary creatinine was measured by akKinetic colorimetric assay (Cobas 8000 analyzer CREA, Roche Diagnostics GmbH). Serum and urinary levels of urea were measured by kinetic assay using Roche automated clinical chemistry analyzers (Cobas 8000 analyzer, UREA/BUN, Roche Diagnostics GmbH). Serum and urinary levels of uric acid were measured by enzymatic colorimetric assay using Roche automated clinical chemistry analyzers (Cobas 8000 analyzer, UA2 Roche Diagnostics GmbH).

##### 10-km—Creatinine, Urea and Uric Acid Were Measured in Serum and Spot Urine

Serum and urinary creatinine were measured by a Kinetic Assay (Alinity c Creatinine Reagent Kit, Abbott Laboratories, Abbot Park, IL, USA). Serum and urinary level of urea were measured by Kinetic Assay on an Abbott automated clinical chemistry analyzers (Alinity c Urea Nitrogen Reagent Kit, Abbott Laboratories). Serum and urinary levels of uric acid were measured by Enzymatic Assay on Abbott automated clinical chemistry analyzers (Alinity c Uric Acid Reagent Kit, Abbott Laboratories).

#### 2.5.3. Measurement of ADMA, SDMA, TMAO 

ADMA, SDMA and TMAO were determined using liquid chromatography coupled with tandem mass spectrometry (LC-MS/MS). Separation was achieved on SeQuant ZIC-HILIC column (50 mm × 2.1 mm, particle size 5 µm) supplied by Merck (Darmstadt, Germany) using an Agilent 1260 Infinity chromatograph (Agilent Technologies, Santa Clara, CA, USA) connected to a hybrid triple quadrupole/linear ion trap mass spectrometer (QTRAP 4000; AB SCIEX, Framingham, MA, USA). Fifty µl of plasma was mixed with the internal standards solution (TMAO-D9 and ADMA-D6) to obtain the final concentration of 0.75 µg mL^−1^ for TMAO-D9 and 155 ng mL^-1^ for ADMA-D6. Next, the samples were mixed on vortex for 5 min, incubated at −20 °C for 20 min and centrifuged at 9300 *× g* in 4 °C for 10 min. The supernatant was transferred to a new test tube, centrifuged at 9300 *× g* in 4°C for 10 min and analysed. The curtain gas, ion source gas 1, ion source gas 2 and collision gas (all high purity nitrogen) were set at 241 kPa, 207 kPa, 345 kPa and “high” instrument units, respectively. The ion spray voltage and source temperature were set at 5500 V and 600 °C, respectively. The chromatographic column was maintained at 40 °C at a flow rate of 0.5 mL min^−1^. The mobile phases consisted of water solution of 20 mM ammonium acetate as eluent A and ACN with 0.2% formic acid as eluent B. The gradient (%B) was as follows: 0 min 90%; 1 min 90%; 7 min 50%; 9 min 50%. The volume of injection was 5 µL. The target compounds were analyzed in multiple reaction monitoring mode. The transitions used for quantitation were *m/z* 203 > 46 and *m/z* 203 > 172 and *m/z* 209 > 77 for ADMA, SDMA and ADMA-D6 and *m/z* 76 > 42 and *m/z* 85 > 46 for TMAO and TMAO-D9, respectively. The compound parameters, viz. declustering potential (DP), collision energy (CE), entrance potential (EP) and collision cell exit potential (CXP) was 61 V, 41 V, 10 V, 0 V and 61 V, 19 V, 10 V, 10 V and 66 V, 45 V, 10 V, 4 V for ADMA, SDMA and ADMA-D6, 66 V, 53 V, 10 V, 6 V and 61 V, 59 V, 10 V, 0 V for TMAO and TMAO-D9, respectively.

#### 2.5.4. Measurement of Albuminuria

Due to technical reasons, creatinine, urea and uric acid were measured after 10- and 100-km races using different methods in the same laboratory. 

##### Albuminuria—100 km

Urinary albumin was measured by an immunoturbidimetric assay (Cobas 8000 analyzer, ALBT2 Roche Diagnostics GmbH).

##### Albuminuria—10 km 

Urinary albumin was measured by an Immunoturbidimetric Assay (Alinity c Microalbumin Reagent Kit, Abbott Laboratories).

#### 2.5.5. Measurement of Urine Human Lipocalin-2/NGAL, TIM-1/KIM-1/HAVCR and Cystatin C Concentrations

Urine concentrations of human lipocalin-2/NGAL, TIM-1/KIM-1/HAVCR and Cystatin C were measured using the immunoenzymatic ELISA method (Quantikine High Sensitivity Human by R&D Systems, Minneapolis, MN, USA) according to manufacturer’s protocol. Minimum detectable concentrations were determined by the manufacturer as 0.012 ng/mL, 0.009 ng/mL, and 0.102 ng/mL, respectively. Intraassay (3.6% for lipocalin-2/NGAL; 4.3% for TIM-1/KIM-1/HAVCR; 6.6% for Cystatin C) and interassay (7.9% for lipocalin-2/NGAL; 6.3% for TIM-1/KIM-1/HAVCR; 7.0% for Cystatin C) respectively. Precisions performances of the assays were determined on 20 replicates from the quality control data of the laboratory. The absorbance were read on the automated plate reader ChroMate 4300 (Awareness Technology, Inc., Palm City, FL, USA) at the wavelength *λ* = 450 nm. The reference curve was prepared according to the manufacturer’s recommendations. 

#### 2.5.6. Calculations and Equations 

Albumin-to-creatinine ratio (ACR) in urine was calculated using the following formula:ACR (mg/g) = urine albumin/urine creatinine (mg/g)(1)

Fractional excretion of urea (FeUrea) and uric acid (FeUA) were calculated using the following formula: Fractional excretion of a parameter (%) = (urine parameter × serum creatinine)/(serum parameter × urine creatinine)(2)

eGFR Chronic Kidney Disease Epidemiology Collaboration Glomerular Filtration Rate Equation (CKD EPI) was calculated using the following equation [31]:eGFR = 141 × min (sCr/κ, 1)^α^ × max (sCr/κ, 1)^−1.209^ × 0.993^age^ × 1.018 [if female](3)
where sCr—serum creatinine; κ is 0.7 for females and 0.9 for males; α is −0.329 for females and −0.411 for males.

##### Acute Kidney Injury Definition and Reference Values for Urinary AKI Biomarkers

Acute kidney injury was defined using AKIN criteria [32]. Stage 1 AKI was defined as a 1.5- to 2.0-fold increase or > 0.3 mg/dL increase in the serum creatinine level from the pre-race creatinine value. Stage 2 AKI was defined as a more than 2- to 3- fold increase in the serum creatinine level. Reference values for urinary NGAL, KIM-1 and Cyst-C according to Pennemans [33] are shown in Table 1. 

### 2.6. Statistical Analysis

We used Statistica 12 software (StatSoft, Kraków, Poland) for the analysis. The Shapiro-Wilk test was applied to assess homogeneity of dispersion from the normal distribution. When the Shapiro-Wilk's test showed a normal distribution, the paired t-test was used. Where the Shapiro-Wilk's test showed that the distributions of the examined parameters were significantly different from normal (*p* < 0.05), the non-parametric Wilcoxon Signed Rank test was used. Descriptive statistics for continuous variables were reported as mean values and standard deviations or median and interquartile range (IQR). The relationships between categorical variables were tested using a chi2 test. Spearman's rank correlation test was used to verify the strength and direction of a relationship between two variables. In all analyses, a *p*-value lower than 0.05 was considered statistically significant. *p*-Values between 0.01 and 0.05, 0.005 and 0.01, 0.001–0.005 were marked as <0.05, <0.01, <0.005, respectively. 

### 2.7. Ethics 

All subjects gave their informed consent for inclusion before they participated in the study. The study was conducted in accordance with the Declaration of Helsinki, and the protocol was approved by the Independent Bioethics Committee of the Medical University of Gdańsk (approval number: NKBBN /434/2018).

## 3. Results

### 3.1. Runners and the Races

Sixteen runners participating in two races were analyzed. All subjects were amateur runners without history of professional running. They had different running experience, with averaged time of regular training of 7.6 ± 6.1 years. Out of 16 runners, 14 completed the 100-km run and two completed 51 km, both distances known as “ultra” runs. All runners completed also a 10-km run. The results of anthropometric analysis, a questionnaire and the results of both races are shown in Table 2. 

### 3.2. Urinalysis 

The incidence of hematuria was higher after the 100-km than 10-km run (*p* < 0.01). The incidence of proteinuria was the same after both races, but the mean level of protein in urine in subjects with proteinuria was higher after the 100-km run than after the 10-km run (*p* < 0.05). Specific gravity increased significantly only after 100-km run. There was also a statistical difference between specific gravity after 10 and after 100 km (*p* < 0.05) (Table 3).

### 3.3. Albuminuria and Urinary New Biomarkers of AKI 

Urinary albumin (uAlb) and albumin-to-creatinine ratio (ACR) increased significantly after both races. ACR above > 200 mg/g (i.e. proteinuria) was found in one runner after the 10-km race and in 2 runners after the 100-km race. All novel AKI biomarkers: uNGAL, uKIM and uCyst-C also increased significantly after both races After the 100-km run, the levels of uAlb, ACR, uNGAL and uCyst-C increased in every runner and uKIM-1 increased in all but two. After the 10-km run, all urinary biomarkers were increased in 10 runners. In remaining 6 runners an increase in at least two studied biomarkers was noticed. The values of studied markers were higher after covering the distance of 100 km than after 10 km, but without statistical significance. All urinary markers were normalized to creatinine. In clinical practice it helps to diagnose AKI in patients with concentrated urine. However, after exercise the creatinine level in urine is dependent on muscle injury. Therefore the interpretation of AKI markers adjusted to urinary creatinine level is problematic. The median values of studied biomarkers are shown in Table 4.

### 3.4. Urea, Uric Acid, Creatinine, eGFR CKD EPI 

The significant increase in all traditional water-soluble uremic toxins were observed after both races; the fractional excretion of urea rose significantly after both races while FeUA was stable (Table 5). Due to technical problem (two different methods used to measure creatinine, urea and uric acid), it was impossible to compare the result after 10 and 100 km.

### 3.5. Novel Water Soluble Toxins 

After both races, SDMA increased significantly. The mean level of SDMA was related to duration of the run and was higher after the 100-km run compared to 10-km run (*p* < 0.005). There were no statistical differences in ADMA and TMAO levels. ADMA was even lower after the 100-km race (Table 6).

Although there was no statistical increase in TMAO in the whole group, the three first runners (all completed the 100-km run within 12 h) had considerably increased TMAO: 1.46-, 3.92- and 2.52-fold (Figure 2).

### 3.6. Relative Changes in Uremic Toxins After Both Races 

After the 10-km race, the highest increase was observed in creatinine level (1.26 fold) and there was no change in urea level. After the 100-km run the highest increases were observed in urea (1.76 fold) and SDMA (1.66 fold) levels. Statistically significant relative changes in the level of studied markers were presented at Figure 3. Correlation between the changes are visualized at Figure 4. It can be observed two groups of biomarkers correlated together, but not between groups. The first one are the novel AKI biomarkers (uNGAL, uKIM-1, uCyst-C), the second include eGFR, Alb, SDMA (not correlated with albumin), creatinine, urea and uric acid. TMAO fold changes were correlated only with changes in creatinine and SDMA. No correlation with ADMA was observed.

Age and gender may influence the level of uremic solutes. In our study no correlation between age and increase of uremic solutes was observed. However, the group included mainly people below 45 years old. Only one person was older and had 57 years. To evaluate the influence of exercise on people in different age separate study should be performed including two subgroups of people also >65 years old. We have not made the analysis on influence of gender on increase in urea, creatinine and uric acid due to low participation of women in the study (*n* = 2).

### 3.7. Correlations between Studied Markers of AKI

There was no significant correlation between novel AKI biomarkers and eGFR. The SDMA levels were correlated with eGFR before (r = −0.64) and after (r = −0.64) the 10-km as well as after 100-km run (r = −0.53). After the 100-km run, there was positively significant correlation between SDMA and urea level (r = 0.62). 

### 3.8. Diagnosis of AKI

Fifty percent of the runners after 100-km and 18% of the runners after the 10-km race fulfilled AKIN criteria of AKI. uNGAL and uCyst-C were slightly above reference values only in isolated cases. uKIM-1 was elevated in 37% of runners after both races. No runner had elevated normalized uNGAL, uKIM-1 and uCyst-C after 100km. After 10km run normalized Cyst-C and KIM-1 were above the reference value in one runner. Data are shown in Table 7.

## 4. Discussion

Man is the best runner among primates [34,35]. During evolution the human kidney, like many other organs, had to adapt to long exercises such as marching or running [34,35]. Nowadays, men run for recreational reasons and pleasure, but our ancestors had to run to find a food and to escape from danger. Therefore, it is hard to understand that running, so natural activity for humankind, could be so harmful for kidneys, leading to AKI in nearly 50% of ultra-marathoners [36]. Running is the most popular sport in the world with millions of enthusiasts [35,36]. For amateur runners, one of the first distances to conquer is 10 km. The 100-km run is one of the most frequently performed ultramarathons, usually organized as a county or mountain race [36]. In presented study, we decided to perform the same analysis concerning kidney function in the same group of runners during two races. Amateurs were chosen to this study for two reasons. First, because the population of running amateurs highly outnumbers professionals and the results from such study are more important for general population and public health. Second, because amateurs are more vulnerable to some injuries due to lack of support from professional trainers, physiotherapist and medical doctors. 

The first findings concerning kidney dysfunction after exercise were found in urine. In 1956, Gardner described hematuria and proteinuria in soccer players and called this condition *“athletic pseudo-nephritis”* [37]. These changes are transient and thought to be benign and not related to any structural changes in kidney. In some cases hematuria is related to bladder trauma or stones [20]. The changes described by Gardner are very typical and were also found in present study. The incidence of hematuria was higher after a 100-km run, but proteinuria was present in the same number of runners. Urine specific gravity was higher after 100 km, which was related to dehydration.

### 4.1. The Changes in Urinary Markers are Universal After Running

Urinalysis is not a quantitative or very sensitive method to establish the protein loss. The albuminuria is more precise. Post-exercise albuminuria was studied in numerous studies after different exercises [20] and was observed even after very short activity [20,38]. Protein loss after exercise is a common functional and benign condition, but its nature and precise mechanism is still unknown [20]. 

There is little information about changes in urinary biomarkers of AKI like NGAL, 1 KIM-1 and Cyst-C after exercise. The AKI biomarkers are very sensitive and established markers of subclinical AKI [8,9,39]. They are elevated in early phase of kidney injury: KIM after 1 h, NGAL after 2 h [9] and uCyst-C after 6 h [40]. KIM-1 is released only from proximal tubular cells [41], NGAL from both proximal and distal tubular cells [41] and uCyst-C is freely filtered in the glomeruli and reabsorbed in the proximal tubule; therefore, almost any presence of uCyst-C in urine reflects tubular dysfunction [40]. The increase in urinary KIM, NGAL and Cyst-C after exercise has been shown in few studies [38,42,43], but the significance of these findings is unknown. It was proposed that an increase in uNGAL may be a metabolic adaptation to endurance exercise or a predisposition to acute kidney injury over time [42]. In presented study, statistically significant increase in urinary excretion of albumin, KIM, NGAL and Cyst-C were found after both races. The values of urinary AKI biomarkers normalized to creatinine level were higher after both races, but only uNGAL normalized to creatinine was significantly increased after both races, although uNGAL/uCr did not exceed normal values in anyone. uKIM/uCr and uCyst-C/uCr were slightly above reference value in only one runner. The importance of these findings is questionable because after exercise urinary creatinine level is dependent not only on urine concentration but also on muscle damage. One of the most interesting findings of this study, worth to emphasize, is that the increase in albuminuria and all AKI markers was universal. These parameters mildly increased almost in every runner after the 100-km run and in the majority of runners after 10 km. It suggest rather physiological than structural changes. On the other hand, only in minority or runners AKI biomarkers reached values typical of AKI. Moreover, even when one marker exceeds the reference values, the other frequently does not. Thus, we suggest that these changes are a physiological reaction to extensive exercise. The increase in AKI markers and albuminuria were observed almost for all patients. It is interesting because post-exercise albuminuria is thought to be functional, not structural, and benign, but new markers were previously related only to structural changes [8]. Further research are needed to verify the observed phenomenon.

### 4.2. Changes in Serum Toxins and Their Fractional Excretion 

The kidney injury leads to decrease in GFR and tubular dysfunction, leading in consequence to accumulation of uremic toxins. The classical markers of AKI are creatinine, urea and uric acid [44]. All these water soluble small compounds are freely filtrated in glomeruli but their later fate in tubules differ a lot. In rest, the creatinine is the best marker because it is practically not reabsorbed or secreted in tubules, while other toxins had more complicated kidney metabolism. Yet, after intense exercise, creatinine is also a marker of muscle injury [45]. In this study, all classical toxins were increased in every runner. The elevation of creatinine and urea was a physiological response to exercise, related to muscle injury (creatinine) and increased metabolism (urea), but rather not to kidney dysfunction. Uric acid changes were different. The mean level increased after both races, but not in every runner. An increase in serum uric acid level is a common finding after exercise and is related mainly to enhanced purine metabolism [46]. Another possible factor contributing to post-exercise hyperuricemia is decreased urinary excretion of uric acid, which was shown after short exercise [47]. In presented study, fractional excretion of uric acid (FeUA) was analyzed and there were no significant changes, although FeUA slightly decreased after the 10-km run and slightly increased after the 100-km race. Similar observation we described previously in marathoners and ultramarathoners [48].

It is unclear why FeUA differs between two races and why kidneys are excreting more UA during very long exercises. There are several factors that can change UA reabsorption in renal tubules during physical activity: dehydration, activation of renin-angiotensin-aldosterone system, vasopressin, and organic anions (e.g. lactate, β-hydroxybutyrate) [49]. The possible explanation of changes in FeUA is toxicity of UA [50]. The decreased UA reabsorption (increased FeUA) during very long physical activity can protect organism from acute hyperuricemia, which could be very harmful [50,51]. In most runners the UA level and FeUA remained stable, which help to minimalize the risk of both hyperuricemia and hyperuricosuria. Due to its invariability, FeUA is not a good marker of renal hypoperfusion. On contrary to FeUA, FeUrea changed a lot after the 100-km race reaching values typical of renal hypoperfusion. 

### 4.3. Changes in New Water Soluble Toxins

The third part of this study was establishing changes in new water soluble uremic toxins: ADMA, SDMA and TMAO. ADMA is an inhibitor of NO synthase (NOS) and a key enzyme for NO production [52,53,54,55]. SDMA is indirectly inhibiting NOS [13,17,55]. NO is the most potent vasorelaxant agent [17,56,57,58] with strong anti-atherogenic, anti-inflammatory and anti-thrombotic action [17,55,58]. Eighty percent of ADMA is degraded by dimethylarginine dimethylamonohydrolase (DDAH), enzyme which is very active in kidney and liver [13,59], and 20% of ADMA in excreted by kidneys [17,52]. SDMA is not metabolized by DDAH and is excreted only by kidneys [54,55]. The levels of ADMA and SDMA are increased in renal failure, but also in subject with cardiovascular disease (CVD) without decreased eGFR [60]. During exercise, the production of NO is increased due to the shear stress in vascular walls [57,61]. Increased levels of NO result in smooth muscle and vascular relaxation [58]. Exercise capacity is positively associated with NO-dependent endothelial function in patients with CVD [54,61]. ADMA decreases blood muscle supply during exercise [54,61]. On the other hand increase of ADMA leads to renin-angiotensin-aldosterone system (RAAS) activation [52], decrease in sodium excretion, and increase in blood pressure [53,59], and all these changes could be beneficial during exercise. There were several studies analyzing the influence of exercise on ADMA after few weeks of training in patients with cardiovascular diseases [53,54,55,56,57,62,63,64]. Most of them showed decrease in ADMA level after series of trainings [53,54,55,56,63,64]. SDMA was analyzed in four studies and its decrease was observed in two studies [53,55]. There were also four studies showing changes in ADMA and SDMA after a single short exercise [17,58,61,65]. In one study, ADMA and SDMA increased [58], in one study decreased [17] and in two studies remained unchanged. All studies were performed in relatively small groups of subjects with cardiovascular diseases. In the presented study, SDMA level increased in every runner after both races. The mean levels of SDMA increased significantly and after the 100-km race were higher than after 10 km. Previous studies confirmed that SDMA is highly correlated with inulin and creatinine clearances [66]. The interesting observation from the presented study is that the SDMA increment was higher than that of creatinine after 100 km (Figure 2). It suggests that SDMA, like urea, is highly reabsorbed in tubules during exercise, probably due to renal hypoperfusion. SDMA levels correlated with urea after the 100-km race. This means that SDMA is not such a perfect marker of GFR in people who are exercising as in the resting state. Unfortunately, fractional excretion of SDMA was not measured in this study.

TMAO is another small water-soluble uremic toxin [67]. It is generated from choline, betaine, and L-carnitine by gut microbial metabolism [68,69]. Under normal physiologic conditions, circulating TMAO is rapidly cleared from the bloodstream, almost exclusively by urinary excretion [69,70]. Renal clearance of TMAO is higher than that of urea and creatinine, which indicates that in addition to glomerular filtration, at least 50% of TMAO renal excretion occurs through tubular secretion [68]. The level of TMAO is strongly correlated with eGFR in CKD [69,70,71]. Plasma level of TMAO is determined also by diet, gut microbial flora and liver flavin monooxygenase activity [72]. Recent reports have shown that circulating levels of TMAO are elevated in several chronic diseases such as obesity, non-alcoholic fatty liver disease, diabetes, heart failure and colon cancer [70,73]. There is a positive correlation between elevated plasma levels of TMAO and an increased risk of major adverse cardiovascular events and death [70,71,72]. Interestingly, it is also suggested that TMAO may directly contribute to progressive renal fibrosis and dysfunction [70,72]. To our best knowledge, TMAO after physical exercise has never been studied. In our study, there was no change in mean TMAO levels. Yet, the interesting observation was that in fastest runners the increase in TMAO was high after the 100-km race—even 3.9-fold. Taking into account that TMAO is a really toxic agent, there is an open question: Is it possible that acute TMAO increase during exercise can cause some harm or diminish physical performance? 

### 4.4. Summary

Fifty percent of runners after 100 km and 18% of runners after the 10-km race fulfilled AKIN criteria of AKI. Almost 40% of the runners exceeded the uKIM-1 reference level in both races, but uNGAL, uCyst-C and all biomarkers normalized to urinary creatinine were within the reference range (almost in all cases). After 100-km race higher incidence of hematuria, higher urine specific gravity and concentration of SDMA were observed than after 10-km race. No significant changes in concentration of uNGAL, uKIM and uCyst-C between races were noted. 100-km race resulted in increased proteinuria, hematuria and urine specific gravity. Urinary albumin, creatinine, urea, uric acid, SDMA, uNGAL, uKIM and uCyst-C increased almost in every runner after 100-km race. After 10-km race the changes was observed less frequently. There were no statistical differences in ADMA and TMAO levels before and after the races. Changes in novel AKI biomarkers were not correlated with nor eGFR neither uremic solutes (both old and novel). Fold change of uremic solutes such as urea, creatinine, uric acid, SDMA correlated with each other. They also correlated with albuminuria (beside SDMA). The significant differences in fold change of markers between 10 km and 100 km were observed for creatinine, urea, uric acid, SDMA, uAlb and eGFR. 

## 5. Conclusions

AKI can be diagnosed on the basis of serum creatinine in a high number of runners after 10- and 100-km races. This diagnosis seems to be highly overestimated due to increased production of creatinine during exercise. Unfortunately, there are no good markers of AKI after exercise. The levels of urea and uric acid are dependent on metabolism. Urea and, probably, SDMA are highly elevated probably due to renal hypoperfusion, a condition that could lead to AKI but is not AKI itself. 

The urinary novel markers of AKI disclose similar changes i.e. increase observed in almost all subjects, such as post exercise albuminuria, and their mild increases seem to be a physiological reaction. To sum up, it is very difficult to diagnose or exclude AKI after long races on the base on classical and even novel markers. We suggest urinary cystatin-C and NGAL as the most promising markers of AKI after exercise. In presented study they were slightly increased in majority of runners but exceed reference value only one runner. On contrary uKIM-1 seems to be too sensitive. uKIM-1 exceeded the reference values almost in 40% of runners both after 10-km and 100-km races, even other markers were within the range. Very recently, Jouffroy et al [74] published a study showing that uNGAL and uKIM-1 progressively increased during ultramarathon, but when normalized to urinary creatinine no statistically significant changes were observed. It seems that our present study is coherent and complementary to this observation. Further studies are needed to establish usefulness of urinary cystatin-C and NGAL to help to distinguish real AKI from “pseudo-AKI”, where no structural changes in kidney are observed, but creatinine values exceeded the references values.

## Figures and Tables

**Figure 1 ijerph-16-04153-f001:**
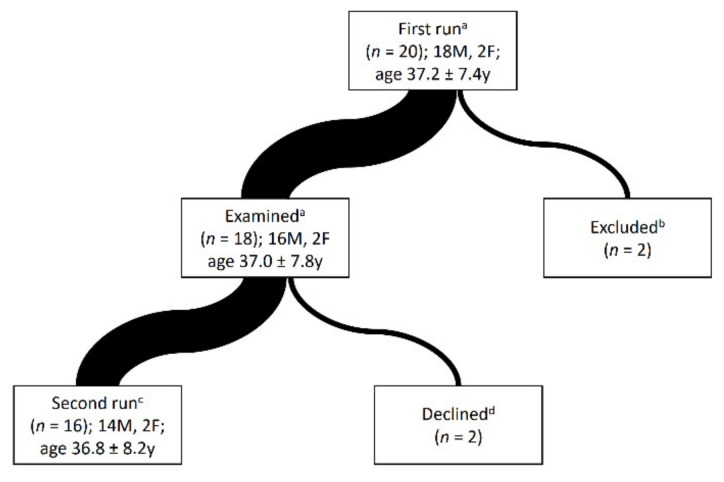
The structure of the study group. ^a^ The blood and urine analyzed; 16 completed the run (100 km); 2 completed half of the run (51 km); ^b^ Two runners were excluded from the study. The first one due to withdrawal after 30 km the second one due to technical problems with blood collection; ^c^ The blood and urine analyzed; all runners completed the run (10 km); ^d^ Two runners did not participate in the second race. Their results from 100 km run were not analyzed.

**Figure 2 ijerph-16-04153-f002:**
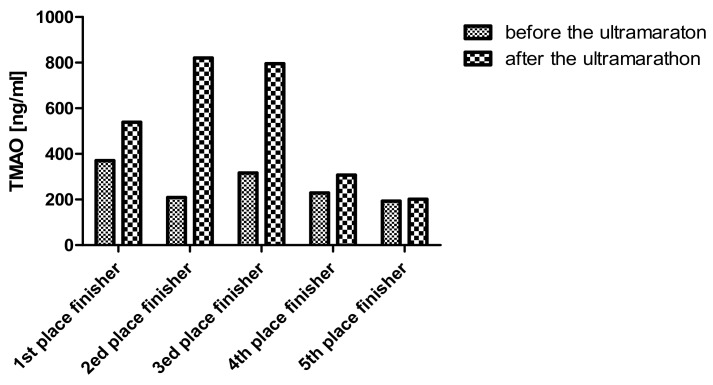
Changes in TMAO in 5 first runners who completed the 100-km race. All the runners were male. Abbreviations: TMAO—trimethylamine N-oxide.

**Figure 3 ijerph-16-04153-f003:**
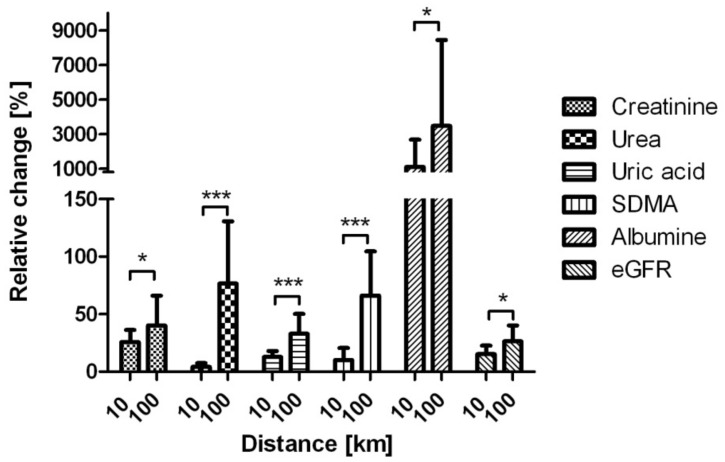
Relative changes [%] in the levels of measured markers between 10-km and 100-km run. Data were presented as mean and standard deviation. Only statistically significant differences were presented: * *p* < 0.05, *** *p* < 0.001. Abbreviations: eGFR—estimated glomerular filtration rate, SDMA—symmetric dimethylarginine,

**Figure 4 ijerph-16-04153-f004:**
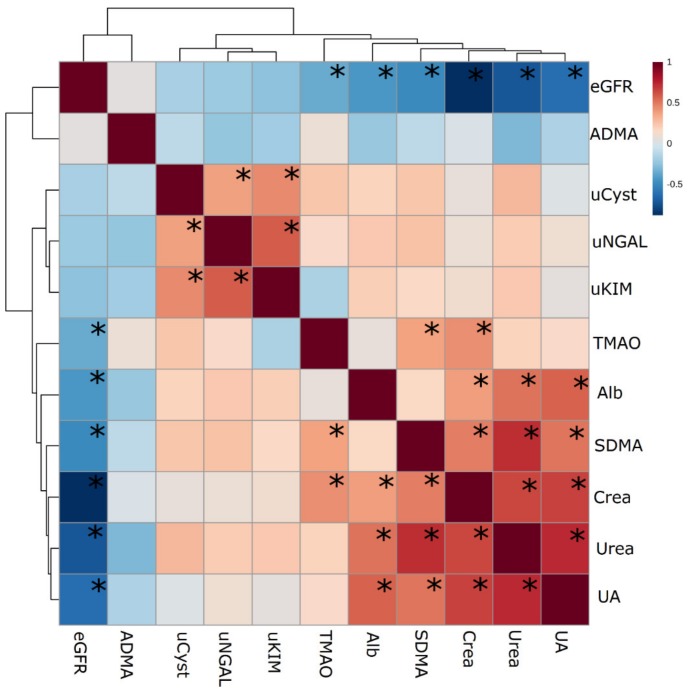
Correlations between the relative level of the toxins and other AKI biomarkers. * Significant correlations (*p* < 0.05); Different colors indicate different Spearman correlation coefficients (strength and direction of correlation) i.e., read color-positive correlation, blue-negative correlation; the darker the color the higher the strength of the relationship. Abbreviations: eGFR-estimated glomerular filtration rate, ADMA-asymmetric dimethylarginine, uCyst-urinary cystatine, uNGAL-urinary neutrophil gelatinase-associated lipocalin, uKIM-kidney injury molecule, TMAO-trimethylamine N-oxide, Alb-urine albumin, SDMA-symmetric dimethylarginine, Crea-creatinine, UA– uric acid.

**Table 1 ijerph-16-04153-t001:** Reference values for urinary NGAL, KIM and Cyst-C [33].

Marker	Reference Values for Healthy Population			
	21–30 Years	31–40 Years	41–50 Years	51–60 Years
uNGAL (ng/mL)	M: < 73.88 F: < 149.26	M: < 87.54 F: < 153.60	M: < 103.95 F: < 158.37	M: < 123.70 F: < 163.62
uNGAL/uCr (μg/g)	M < 125.5 F < 243.2	M < 122.1 F < 236.5	M < 127.6 F < 247.0	M < 142.9 F 276.6
uKIM-1 (ng/mL)	M: < 1.86 F: <1.56	M: < 2.06 F: < 1.74	M: < 2.28 F: < 1.94	M: < 2.52 F: < 2.15
uKIM/uCr (μg/g) both sexes	< 2.28	< 2.14	< 2.24	< 2.61
uCyst-C (ng/mL)	M: < 208.2 F: < 180.4			
uCyst/uCr (μg/g) both sexes	< 220	< 191	< 183	< 193

*Abbreviations*: M—male, F—female, uNGAL—urinary neutrophil gelatinase-associated lipocalin, uNGAL/uCr -urinary NGAL to creatinine ratio, uKIM-1—kidney injury molecule 1, uKIM/uCr- urinary KIM-1 to creatinine ratio, uCyst-C—urinary cystatin C, uCyst/uCr -urinary cystatine C to creatinine ratio.

**Table 2 ijerph-16-04153-t002:** General characteristics of the study group.

Variable	Result
Age (years)	36.8 ± 8.2
Weight (kg)	74 ± 13
Height (cm)	175.9 ± 8.7
BMI (kg/m^2^)	23.8 ± 3.1
WHR	0.8 ± 0.05
The body fat percentage (%)	13.4 ± 4.9
The mean heart rate (bpm)	56 ± 11
Duration of regular running (years)	7.6 ± 6.1
Mean training (days/week)	5.5 ± 1.7
Average training (km/month)	146 (112–265)
**Results of ultra-marathon**	
15 runners finished 100 km 2 runners finished 51 km	13 h 33 min ± 01 h 55 min 7 h 55 min and 7 h 56 min
**Results of 10-km race**	45 min 27 sec ± 4 min 16 sec

Data were presented as mean ± SD, or median (IQR); Abbreviations: BMI—body mass index; WHR—waist-to-hip ratio; bpm—beats per minute.

**Table 3 ijerph-16-04153-t003:** Changes in urinalysis after both races.

Marker	10 km			100 km		
	Before	After	*p*	Before	After	*p*
Specific gravity	1014 ± 6	1015 ±6	ns	1015 ± 8	1023±7	<0.005
pH	6.2 ±1.1	6.2 ±1.1	ns	5.84 ± 0.85	5.5±1.0	ns
Hematuria ^a^ (*n*)	0/16	1/16	ns	0/16	8/16	<0.005
Overt proteinuria ^b^ (*n*)	0/16	9/16	<0.001	0/16	8/16	<0.005
Protein in urine ^c^ (g/l)	na	0.37 (0.25–0.43)	-	na	0.72 (0.54–0.97)	-

Data are presented as mean ± SD, or median (IQR), ns-non significant, *p* > 0.05, na-not applicable, no proteinuria in strip test was detected before the runs, ^a^ hematuria was defined as: > 2 red cells under high-power field of view; ^b^ proteinuria in strip test; ^c^ mean level of protein in urine in subjects with proteinuria in strip test.

**Table 4 ijerph-16-04153-t004:** Changes in new urinary AKI biomarkers after both races.

Marker	10 km			100 km		
	Before	After	*p*	Before	After	*p*
uAlb (mg/L)	5.7(5.0–7.6)	41(16–126)	<0.001	3.0(3.0–9.5)	57(27–306)	<0.0005
ACR (mg/g)	9(5.2–14.20)	31(11–81)	<0.005	7.0(3.2–15)	30(14–94)	<0.005
uNGAL (ng/mL)	4.9(0.8–8.3)	19(9.0–30)	<0.005	4.4(0.3–11)	30(19–63)	<0.0005
uNGAL/uCr (μg/g)	4.09(1.92–6.35)	11.89(9.28–18.15)	<0.005	4.56(2.97–6.98)	13.42(9.92–26.49)	<0.005
uKIM-1 (ng/mL)	0.44 (0.15–1.1)	1.3(0.19–2.5)	<0.01	0.21(0.09–0.78)	1.6(0.58–2.5)	<0.001
uKIM/uCr (μg/g)	0.49(0.34–0.81)	0.72(0.34–1.45)	ns	0.37(0.07–0.99)	0.68(0.35-1.05)	ns
uCyst-C (ng/mL)	46 (5.7–90)	118(72–144)	<0.005	28(2.4–88)	139(116–156)	<0.0005
uCyst/uCr (μg/g)	49.56 (11.49–72.03)	68.88 (52.96–96.77)	<0.05	34.54 (6.15–73.38)	67.31 (48.36–75.78)	ns (*p* = 0.08)

Data are presented as median (IQR), ns-non significant, *p* > 0.05; Abbreviations: uAlb – urinary albumin, ACR—albumin-to-creatinine ratio, uNGAL—urinary neutrophil gelatinase-associated lipocalin, uNGAL/uCr -urinary NGAL to creatinine ratio, uKIM-1—kidney injury molecule 1, uKIM/uCr- urinary KIM-1 to creatinine ratio, uCyst-C—urinary cystatin C, uCyst/uCr -urinary cystatine C to creatinine ratio.

**Table 5 ijerph-16-04153-t005:** Change in basic uremic toxins after both races.

Markers	10 km			100 km		
	Before	After	*p*	Before	After	*p*
Creatinine ^a^ (mg/dL)	0.79 ± 0.14	0.99 ± 0.18	<0.001	0.85 ± 0.13	1.19 ± 0.26	<0.001
eGFR CKD-EPI (mL/min/1.73m^2^)	112 ± 13	95 ± 15	<0.001	107 ± 13	78 ± 15	<0.001
Urea ^a^(mg/dL)	32(31–39)	33 (32–42)	<0.005	34(30–37)	66 (49–70)	<0.001
FeUrea (%)	44 ± 12	3314	<0.01	49 ± 12	29 ± 11	<0.001
Uric acid ^a^(mg/dL)	5.03 ± 0.99	5.65 ± 0.06	<0.001	4.22 ± 0.76	5.6 ± 1.2	<0.001
FeUA (%)	6.0 ± 1.6	5.46 ± 1.95	ns	5.59 ± 1.6	6.1 ± 2.3	ns

Data are presented as mean ± SD, or median (IQR), ns-non significant, *p* > 0.05; ^a^ two different methods were used to measure creatinine, urea and uric acid; Abbreviations: eGFR CKD-EPI—estimated glomerular filtration rate using CKD-EPI (Chronic Kidney Disease Epidemiology Collaboration) equation; FeUrea—fractional excretion of urea, Fe UA -fractional excretion of uric acid.

**Table 6 ijerph-16-04153-t006:** Changes in ADMA, SDMA and TMAO.

Markers	10 km			100 km		
	Before	After	*p*	Before	After	*p*
ADMA (ng/mL)	133 ± 25	137 ± 17	ns	136 ± 16	123 ± 24	ns
SDMA (ng/mL)	287 ± 51	312 ± 38	<0.01	227 ± 41	369 ± 79	<0.001
TMAO (ng/mL)	329 (212–502)	429(243–520)	ns (*p* = 0.07)	306 (225–447)	513 (262–751)	ns

Data are presented as mean ± SD, or median (IQR), ns-non significant, *p* > 0.05; Abbreviations: ADMA—asymmetric dimethylarginine, SDMA—symmetric dimethylarginine, TMAO—trimethylamine N-oxide.

**Table 7 ijerph-16-04153-t007:** Number of runners with values typical for AKI.

Subjects	AKI Stage 1 ^a^	AKI stage 2 ^a^	uNGAL above ref. value ^b^	uKIM-1 above ref. value ^b^	uCyst-C above ref. value ^b^	uNGAL/uCr above ref. value ^b^	uKIM/uCr above ref. value ^b^	uCyst/uCr above ref. value ^b^
10-km run	3	0	1	6	1	0	1	1
100-km run	7	1	1	6	0	0	0	0

^a^ According to AKIN criteria; ^b^ According to Pennemans [33]; Abbreviations: AKI–acute kidney injury, eGFR CKD-EPI–estimated glomerular filtration rate using CKD-EPI (Chronic Kidney Disease Epidemiology Collaboration) equation; ACR–albumin-to-creatinine ratio; uNGAL–urinary neutrophil gelatinase-associated lipocalin, uKIM-1–kidney injury molecule 1, uCyst-C–urinary cystatin C., uNGAL/uCr-urinary NGAL to creatinine ratio, uKIM/uCrurinary KIM-1 to creatinine ratio, uCyst/uCr -urinary cystatine C to creatinine ratio.

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
