# Peer review of "Changes in Water Soluble Uremic Toxins and Urinary Acute Kidney Injury Biomarkers After 10- and 100-km Runs"

_ijerph, 2019, doi:10.3390/ijerph16214153_

Round 1

Reviewer 1 Report

The paper is well written, and their work is well presented. It should be published after minor and major revisions shown below.

Minor points

-Modify figure 1 is not visible

-Change table 4 to sheet 9

-Homogenize references some have DOI and others do not.

Major points

-In Figure 2 specify if it is a female or male person, discuss the prevalence of gender in the TMAO levels in the first 5 runners.

-Figure 4 is missing information about colors

-In the discussion part, changes in uremic toxins do not speak to about age and gender if these influence the increase in urea, creatinine and uric acid. Please discuss.

-In the materials and methods part specify the exclusion and inclusion criteria for patient selection.

Mention if there are runners of different races and how it affects the results of AKI.

Author Response

Dear reviewer
We appreciate your dedicated efforts and thoughtful consideration in reviewing this article. We hope that the corrections will satisfy Reviewers and that the current revised version of our manuscript will be acceptable for publication in this journal.

1.Modify figure 1 is not visible
Answer: The figure was modified. We hope that the modifications increased its visibility.

Change table 4 to sheet 9
Answer: Could you kindly explain what changes should be made with Table 4?
Probably the problem was with title of table 2, I was changed. Homogenize references some have DOI and others do not.
Answer: DOI numbers were removed In Figure 2 specify if it is a female or male person, discuss the prevalence of gender in the TMAO levels in the first 5 runners.
Answer: All 5 first runners were male. Thus no comparison can be made. We added this information in figure caption. Figure 4 is missing information about colors
Answer:
Thank you for that comment. The appropriate explanation was added. In the discussion part, changes in uremic toxins do not speak to about age and gender if these influence the increase in urea, creatinine and uric acid. Please discuss.
Answer. The influence of age on increase of uremic solutes after exercise can be very interesting and another study aimed only on the subject should be performed. Our group was rather homogenous regarding the age. Thus, no conclusion can be drawn. The appropriate information was added. We have not made the analysis on influence of gender on increase in urea, creatinine and uric acid due to low parcitipation of women in the study (n=2) In the materials and methods part specify the exclusion and inclusion criteria for patient selection.
Answer. The exclusion and inclusion criteria can be found in Materials and Methods section. However, it was hard to find the subsection “Subjects selection” was created.

8.. Mention if there are runners of different races and how it affects the results of AKI.
Answer: All the runners were of Caucasian race. The appropriate information was added to the text.

Reviewer 2 Report

The authors reported the difference of kidney injury biomarkers after run. Several markers increased with 10 or 100 km run, and to assess uremic toxins comprehensively after run will be important to understand exercise-induced acute kidney injury.

The authors discussed AKIN or KDIGO guideline only about the change of serum creatinine level, but not decrease of urine output. The change of urine volume is critical for exercise-induced acute kidney injury, and they should mention it in the Introduction or Discussion.

Table 1 should be put in the Result. Do they have data of change of physical parameters, such as body weight and blood pressure, before and after run?

The authors should show urine creatinine level, and also adjust urine markers with urine creatinine because urine after run should be concentrated.

Author Response

Dear reviewer
We appreciate your dedicated efforts and thoughtful consideration in reviewing this article. We hope that the corrections will satisfy Reviewers and that the current revised version of our manuscript will be acceptable for publication in this journal.

1.The authors discussed AKIN or KDIGO guideline only about the change of serum creatinine level, but not decrease of urine output. The change of urine volume is critical for exercise-induced acute kidney injury, and they should mention it in the Introduction or Discussion.
Answer: The second criterium used to diagnose AKI according to AKIN is decreased urine output. It is very difficult to precisely assess urine output without bladder catheterization and without prolong (at least 6 hours) observation, therefore it cannot be used in running subjects. The appropriate explanation was added.

2.Table 1 should be put in the Result. Do they have data of change of physical parameters, such as body weight and blood pressure, before and after run?
Answer The Table 1 was moved to the results (now, as a table 2). Due to technical problems no evaluation of changes of weight and blood pressure before and after run was performed.

3.the authors should show urine creatinine level, and also adjust urine markers with urine creatinine because urine after run should be concentrated.
Answer: All urinary markers were normalized to creatinine. In clinical practice it helps to diagnose AKI in patients with concentrated urine. However, after exercise the creatinine level in urine is dependent on muscle injury. Therefore the interpretation of AKI markers adjusted to urinary creatinine level is problematic. We add the appropriate data to results.

Reviewer 3 Report

The topic is interesting since more and more people practice exercise such as marathon and ultra-trail without support from professional trainers, exercise which is considered now to be unhealthy. In addition, the manuscript is well written and documented.

Major revisions:

1) However, I think it will be more appropriate to express results normalized with urinary creatinine for uNGAL, uKIM-1 and uCyst-C.

2) Other works find no difference for the new markers proposed here, uNGAL (for example, "Acute kidney injury during an ultra-distance race. Jouffroy R et al 2019 14(9):e0222544 »). The authors have to discuss this point.

Minor revisions:

In some references, DOI number have to be removed.

Author Response

Dear reviewer
We appreciate your dedicated efforts and thoughtful consideration in reviewing this article. We hope that the corrections will satisfy Reviewers and that the current revised version of our manuscript will be acceptable for publication in this journal.

Major revisions:

1). However, I think it will be more appropriate to express results normalized with urinary creatinine for uNGAL, uKIM-1 and uCyst-C.
Answer:
All urinary markers were normalized to creatinine. In clinical practice it helps to diagnose AKI in patients with concentrated urine. However, after exercise the creatinine level in urine is dependent on muscle injury. Therefore the interpretation of AKI markers adjusted to urinary creatinine level is problematic. We add the appropriate data to results.

2). Other works find no difference for the new markers proposed here, uNGAL (for example, Acute kidney injury during an ultra-distance race. Jouffroy R et al 2019 14(9):e0222544 »). The authors have to discuss this point.

Answer: Jouffroy et al, very recently (4 days before we submitted our manuscript) published a study showing that uNGAK and uKIM-1 progressively increased during ultramarathon, but when normalized to urinary creatinine no statistically significant changes were observed. It seems that our present study is coherent and complementary to Jouffroy study. We added this information to the text

In some references, DOI number have to be removed.
Answer: DOI numbers were removed

Round 2

Reviewer 2 Report

The authors responded to the reviewer’s comments. The reviewer has understood that they discussed well about some points even they did not have enough data. There are two minor points as bellow.

It is possible that urine volume is measured without bladder catheter in the clinical situations, and the authors should discard the need of bladder catheter to check urine volume (Line 80 and 81).

In the Conclusion, the authors discussed the urine markers with/without urine creatinine adjustment, but they should put it in ‘4.1. The change in urinary markers are universal after running (Line 406), and discuss it more deeply.

Author Response

Dear reviewer

Changes in urine output are very important in diagnosis of kidney function, but it is very difficult to measure it during exercise. We performed such a study during 100km run performed on the 400m track (Wołyniec W., et al. Glomerular Filtration Rate Is Unchanged by Ultramarathon. J Strength Cond Res. 2018;32(11):3207-3215) and therefore we know how difficult and time consuming it is. It is also impossible to perform such a study and to collect all urine during trail run.

The text in introduction was changed:

“There are two main problems with diagnosis of AKI based on urine output in exercising subjects. First, it is very difficult to precisely assess urine output during ultramarathon. Subject running 100km in forest are urinating several times in different places. There is no chance to collect urine in such a race. Second, according to AKIN the prolog, at least 6 hours, observation is needed. It could be easily done in hospital, but it is very difficult to observe marathoners who completed 100km run after finish line for so long. Therefore in most studies a diagnosis of AKI after ultramarathon is based only on laboratory tests”.

2.

The discussion concerning  normalized values of AKI biomarkers was added:

“The values of urinary AKI biomarkers normalized to creatinine level were higher after both races, but only uNGAL normalized to creatinine was significantly increased after both races, although uNGAL/uCr did not exceed normal values in anyone. uKIM/uCr and uCyst-C/uCr were slightly above reference value in only one runner. The importance of these findings is questionable because after exercise urinary creatinine level is dependent not only on urine concentration but also on muscle damage.

Reviewer 3 Report

The changes made by the authors have been correctly discussed.  This topic is interesting since more and more people are running and it will be important to better characterize injuries induced by running. 

Author Response

Dear Reviewer

Thank you for your comments